# African Swine Fever: A Review of Current Disease Management Strategies and Risks Associated with Exhibition Swine in the United States

**DOI:** 10.3390/ani13233713

**Published:** 2023-11-30

**Authors:** Hannah J. Cochran, Angela M. Bosco-Lauth, Franklyn B. Garry, I. Noa Roman-Muniz, Jennifer N. Martin

**Affiliations:** 1Department of Animal Sciences, Colorado State University, Fort Collins, CO 80521, USA; hannah.cochran@colostate.edu (H.J.C.); noa.roman-muniz@colostate.edu (I.N.R.-M.); 2Department of Biomedical Sciences, Colorado State University, Fort Collins, CO 80521, USA; angela.bosco-lauth@colostate.edu; 3Department of Clinical Sciences, Colorado State University, Fort Collins, CO 80521, USA; franklyn.garry@colostate.edu

**Keywords:** African swine fever, exhibition swine, biosecurity, infectious disease, foreign animal disease

## Abstract

**Simple Summary:**

As African swine fever approaches the United States, it is important that the swine industry has safeguards in place to prevent catastrophic disruptions in the event of an outbreak. This review outlines current knowledge of African swine fever, disease management strategies, and the risks presented by the unique exhibition swine sector.

**Abstract:**

African swine fever is a high-consequence foreign animal disease endemic to sub-Saharan Africa and the island of Sardinia. The U.S. is the world’s third largest pork producer, and ASF introduction would severely disrupt the pork supply chain, emphasizing the need to protect market access for U.S. proteins. However, niche producers raising swine intended for exhibition may not follow stringent biosecurity protocols, and livestock show circuits may promote untracked animal movement across the country, potentially exacerbating virus’ spread in the event of ASF incursion into the U.S. Youth membership in state or national swine organizations offers a route for outreach and educational activities to enhance foreign animal disease preparedness, and adult presence at swine exhibitions allows a wide variety of programming for all ages to better serve all levels of understanding.

## 1. Introduction to African Swine Fever

### 1.1. Characteristics: Endemicity

African swine fever (ASF), a high-consequence, reportable hemorrhagic fever caused by African swine fever virus (ASFV), results in high mortality rates in infected swine. ASFV is a double-stranded DNA virus first described in Kenya in 1921 and the only member of the *Asfarviridae* family. There are vaccines offered in Vietnam and the Philippines; however, there are no commercially available vaccines in the United States [1]. Thailand and North Macedonia reported their first occurrences of ASF in January 2022, and two countries with outbreaks prior to 2007 have successfully eradicated the virus: Belgium and Czechia [2]. It is not currently found in the United States, but it is endemic to sub-Saharan Africa, Eastern Europe, and the island of Sardinia [3]. With endemicity in such a variety of regions and other countries exhausting all efforts to prevent the virus from crossing their borders, close contacts between humans and animals have the potential to exacerbate disease spread throughout the world via fomites and reintroduce diseases into countries where they were previously eradicated as globalization bolsters trade [4]. Prior to recent outbreaks, de Glanville et al. (2014) assessed the virus’s potential for endemicity in Africa using spatial multi-criteria decision analysis (MCDA). Their analyses indicated the suitability for endemicity due to domestic swine transmission in much of sub-Saharan Africa, while other regions showed suitability for endemicity via wildlife reservoirs. These algorithms accounted for domestic pig population density, *Ornithodoros* tick spp. prevalence, and wild hog populations, including warthogs, bushpigs, and giant forest hogs. Domestic transmission cycles were supported by proximity to trade, tick populations, and pig density. Wildlife transmission cycles were supported by tick populations, pig density, and habitat suitability for wild hogs. Most of Northern Africa is unsuitable for ASF introduction from either pathway, but the potential for spillover from wildlife should be monitored [5].

### 1.2. Characteristics: Transmission and Detection

Understanding disease transmission pathways, as well as familiarity with disease detection strategies, is of paramount importance in foreign animal disease mitigation. In the event of an ASF outbreak in the United States, the greatest concern is transmission to the feral swine population. These pigs are escaped domestic swine, Eurasian wild boars introduced by early settlers, and hybrids of both species. The United States Department of Agriculture Animal and Plant Health Inspection Service (USDA APHIS) estimates the feral swine population at 6 million, and it is rapidly growing [6]. Feral swine have been reported in 35 states [7], and with no natural predators and limited human control, they represent a growing threat to U.S. agriculture and livestock.

Feral swine are known reservoirs of diseases, including brucellosis, porcine reproductive and respiratory syndrome (PRRS), and pseudorabies. Domestic swine raised in commercial, vertically integrated facilities have minimal contact with wildlife, as these producers follow strict biosecurity protocols. However, hobby and niche swine producers, such as those who raise animals for exhibition, often do not follow these biosecurity protocols, so animals may have the opportunity to encounter wildlife. In a 2009 study, feral swine in Texas were found to cross 100- and 500-m buffer zones around domestic swine pens, particularly those housing females. Feral–domestic contact could be mitigated via the construction of appropriate fencing around swine facilities [8]. However, such measures are often cost prohibitive.

Transmission of ASF most commonly occurs through direct contact with infected animals, tick bites, and the consumption of contaminated swill. The virus remains stable in many substrates, including feed and soil. In Vietnam, with ongoing ASF outbreaks, truck cabs, employee clothing, and surfaces in high-traffic areas were identified as the greatest sources of positive samples in the feed mill of a vertically integrated swine production facility. After the mill implemented extensive disinfection procedures and restricted drivers from exiting their cabs, positivity was reduced in later sampling [9]. The identification of such biosecurity gaps is crucial in modern pork production systems.

ASFV Georgia 2007/1 was used to study the half-life of the virus in feed ingredients over a simulated 30-day trans-Atlantic shipment. Samples were collected at 1, 8, 17, and 30 days after inoculation, and ASFV was quantified via titration. The half-life in feed ingredients ranged from 9.6 to 14.2 days compared to 8.3 days in viral growth media, supporting the theory that feed ingredients increase viral stability. This study demonstrates that the transport of contaminated feed products is a viable pathway for ASF to spread between countries [10]. Some swine disease outbreaks, including a 2014 Ohio porcine epidemic diarrhea virus (PEDV) epizootic, have been traced back to feed products. The Ohio outbreak was tied to contaminated feed pellets from a new supplier for the breed-to-wean production flow, resulting in the 100% mortality of the affected piglets. The control and surveillance of feed and related products prior to import to the United States or introduction to production facilities could prevent such outbreaks in the future. Significant research highlighting the risk of survival and transmission in feed and related products is available and should inform any efforts aimed at reducing the risk of ASFV transmission in swine populations.

Treating patches of soil where deceased wild boars are found with no readily evident cause of death (e.g., hit by vehicle or gunshot) may become important in countries where ASF is endemic, as Carlson et al. (2020) found that the virus may survive for up to two weeks depending on the soil type. Three soil types were exposed to blood from an ASF-positive wild boar: beach sand, forest soil, and swamp mud. Beach sand showed high initial titers, but no infectious virus was recovered after three days. No infectious virus was recovered from the acidic forest soil even immediately after treatment, and the swamp mud had low titers initially but declined after three days. However, the viral genome could be detected in all soil types throughout the entire observation period [11].

Regarding measures taken to limit the spread of ASF throughout the European Union, the EU Commission publishes minimum guidelines for biosecurity plans. These guidelines include requiring ASF-free certification for the movement of semen and ova or the acquisition of new animals, visitor restrictions, and training for workers [12]. Additionally, when hunting in regions where the virus is endemic or has a current outbreak, the hunters must be trained in basic biosecurity protocols, evisceration is to be carried out with gloves and hands should be washed afterwards, any clothing worn during the hunt should be washed at high temperatures, and the hunters should avoid contact with domestic swine for 48 h [13].

Swine experimentally infected with ASFV strains of varying virulence showed clinical signs of disease and seroconversion (with low virulence strain) prior to euthanasia. Animals were euthanized between 3- and 26-days post-infection, and samples were collected from multiple tissues and muscles. Frozen muscle samples were thawed prior to use and gently squeezed to extract exudate. Polymerase chain reaction (PCR) was performed on whole-blood, muscle, and tissue samples, while ELISAs were run on serum samples. Cycle threshold (Ct) values from PCR closely correlated with ELISA results, with highly and moderately virulent strains resulting in lower Cts. Meat sample Ct values trended lower than the corresponding blood samples, and no genomic material was found in meat samples from pigs infected with the low virulence strain. The presence of ASFV genomic material in meat exudate likely corresponds to virulence, and in the absence of gold-standard serum or whole-blood samples, meat products such as those carried by airline passengers may be used to test for ASFV [14].

The introduction of zoonoses or foreign animal diseases (FADs) into the United States via animal products carried by travelers is a real risk. On average, 8000 units of pork products are confiscated by the Customs and Border Patrol (CBP) at U.S. ports of entry yearly [15]. The sanitary status of these products is unknown, and they could easily carry diseases unbeknownst to the passenger. However, the rate of detection is typically under 50%, allowing the entry of a large quantity of pork products daily. Jurado et al. (2018) built two statistical models to estimate the monthly probability of ASFV entry into the country. These models accounted for the number of pork products confiscated by the CBP at airports and the number of passengers arriving in the U.S. via airplane at 87 international airports. No data were available regarding the exact country of origin of the pork products. For the probability of ASFV entry, the high-risk airports were Dulles, JFK, Houston-Bush, Warwick, and San Juan, with Ghana, Cape Verde, Ethiopia, and Russia being the highest-risk countries. May and July, two high-traffic summer months, carried the highest risk. Baggage inspection is limited by workforce and inspection times, and not every piece of luggage is inspected. Naturally, the risk of introduction increases during summer months in line with travel volume [15].

Currently, validated testing methods for ASF detection requires animal restraint, such as nasal swabs and blood draws, or involves postmortem tissue collection, as with spleen and tonsil samples. Oral fluid testing is often used to detect other swine viral pathogens, including influenza A virus (IAV), PRRSV, and porcine epidemic diarrhea virus (PEDV). Cotton ropes are hung in pens at shoulder height, and animals are allowed to chew on them for 20–60 min, depending on their prior experience with rope testing. After the collection period, fluid is wrung out and stored for testing [16]. Experiments seeking to validate the use of oral fluids for ASFV detection were carried out at Plum Island Animal Disease Center (PIADC, Orient, NY, USA) and the National Centre for Foreign Animal Disease (NCFAD, Winnipeg, Manitoba, Canada), utilizing one animal per pen as a “seeder pig.” This pig was inoculated with ASF Georgia 2007/1 (highly virulent) or ASF Malta ’78 (moderately virulent) and housed in a pen with 19–24 naïve pigs. Beginning on day 0 post-infection and continuing throughout the duration of this study, oral fluid ropes were placed in the pen and chewed for up to half an hour, before being wrung out. ASFV genomic material was found in individual samples as early as 3 days post-infection and on every subsequent day until termination of the study as animals succumbed to disease. Research in Spain used ASFV NHV/P68 (NHV), a low-virulence strain, to develop a recombinant protein ELISA and successfully recovered antibodies from oral fluid samples [17]. Further research into the viability of oral fluids as a testing method for ASF is crucial, as current methods of individual pig sampling are labor and resource intensive, while oral fluid is an aggregate sample that requires minimal manpower and fewer laboratory resources [18].

### 1.3. Management: Disinfection and Survivability

Present ASFV management strategies are limited, as the mortality rate is high and there is no treatment. The disinfection of hard surfaces, such as pens and trailers, and decontamination of possible infectious materials, including soil and pork products, are the best methods of prevention. When beach sand and potting soil samples inoculated with infectious blood were treated with calcium hydroxide or citric acid, no virus was recovered after one hour of decontamination [11]. The application of lime as a disinfectant has also been effective in vitro, with three lime products showing a reduction in viral titer when applied to six German soil types. A 10% solution of either lime milk, slaked lime, or quicklime was sufficient to inactivate ASFV, regardless of the water content of the soil. Additionally, previous studies demonstrated the efficacy of 0.1% peracetic acid in completely inactivating ASFV in all tested soil types, although it was limited by the presence of blood [19]. Lime was unaffected by the presence of blood, indicating its usefulness in areas surrounding infectious carcasses that are likely to be tainted with blood and other bodily fluids. The appropriate decontamination of the soil where infected animals died is critical in curbing the further spread of ASFV, especially in countries such as Germany, where the virus has spilled over into the wild boar population [20].

Multiple methods of inactivation have been tested, including the use of a formaldehyde-based product (FBP) on ASFV in cell culture. ASFV titers were reduced by four logs in vitro after the application of four concentrations of the FBP, although this does not necessarily reflect its efficacy in vivo, such as in commercial feed-manufacturing settings [21]. When applied to 10^7^ and 10^5^ TCID_50_ ASFV, highly complexed 5% iodine demonstrated the significant inactivation of ASFV after five minutes of immersion or spray treatment compared to commercially available 5% povidone-iodine, which did not inactivate ASFV until 15 min of immersion. Highly complexed 5% iodine, therefore, has the potential for use in industrial applications as a rapid disinfectant [22]. Some commercially available disinfectants, including the liquid Virkon^TM^ S have been tested against ASFV on surfaces commonly found in production facilities, such as stainless steel and concrete. The application of 1% Virkon^TM^ S to inoculated concrete and stainless-steel sections resulted in the complete inactivation of ASFV after a 10-minute contact time. Additional compounds tested in cell culture include sodium hypochlorite, glutaraldehyde, caustic soda, potassium peroxymonosulfate, phenol, acetic acid, and benzalkonium chloride. All seven of these compounds effectively inactivated ASFV at multiple concentrations recommended by the World Organisation for Animal Health (WOAH), although some were affected by the presence of organic material in highly soiled conditions [23], demonstrating the importance of cleaning surfaces prior to disinfection.

Ozone is commonly utilized in the food and medical industries as a disinfectant. It quickly sterilizes surfaces and produces oxygen as a decomposition product, and it has inactivated RNA viruses in previous studies, with little work performed to study its effects on DNA viruses. Porcine alveolar macrophages were inoculated with wild-type ASFV and titered to ensure infectivity. ASFV isolates were incubated with ozonized water produced via an electrolytic ozone generator for 1, 3, 6, or 10 min prior to inoculation back into cell culture. Following additional titration, it was determined that ozonized water reduced the concentration of ASFV by at least two logs at all time points assessed within the study [24].

Prior research has shown that commercial meat curing and drying processes can inactivate multiple viruses. Although swine presenting with severe clinical signs at a slaughter plant would be prohibited from slaughter by the USDA after antemortem inspection, it is necessary to use these animals for processed meat products in this study to ensure the virus’ presence in the selected tissues. Infectivity titers from inoculated live pigs ranged from 5.4 to 9.5 −log_10_/mL, depending on the tissue. Shoulders, hams, and loins were cured according to traditional Iberian or Serrano methods, and samples were collected throughout the process. ASFV was inactivated after 140 days in all cuts, indicating that the Iberian and Serrano curing processes destroy the virus [25].

Studies examining the survival (i.e., ability of the virus to withstand lethality treatments) of ASFV have shown that the virus is capable of surviving in various substrates for extended times at 4 °C, including an estimated 112.6 days in hay and 97.4 days in straw. Leaf litter offered the lowest survival time at 18.9 days, while water and soil may sustain the virus for 35.6 and 32.8 days, respectively. This study illustrates that contaminated surfaces, such as those where wild boar carcasses are recovered, may contribute to virus survival, depending on the substrate. Additionally, ambient temperatures of around 4 °C will allow ASFV to survive for longer periods of time in certain substrates [26].

### 1.4. Management: Vaccines and Emerging Candidates

African swine fever control, at present, relies heavily on the destruction of all pigs inside of control zones and monitoring surrounding areas. In the summer of 2023, commercially available live-attenuated vaccines were licensed in the Philippines and Vietnam. Previous studies evaluated inactivated viruses as vaccine candidates and found a lack of protective immune response when animals were challenged with virulent isolates [27,28,29,30]. More recently, attention has turned to live-attenuated vaccines, although concerns about reversion remain to be addressed. Multiple vaccine candidates are in progress, representing years of research across several countries, with few licensed vaccines.

One commercially available vaccine, namely NAVET-ASFVAC, utilizes the deletion of gene I177L from the highly virulent ASFV strain Georgia. This previously undiscovered gene is conserved across multiple ASFV strains isolated from different countries, and animals inoculated with ASFV-G-ΔI177L did not develop clinical signs, remaining healthy during the 28-day observation period compared to animals inoculated with the parental Eurasia strain [31]. Further research into the safety of this strain showed minimal reversion when naïve animals were exposed to inoculated animals, with the vaccine strain remaining phenotypically attenuated [32]. The vaccine developed from the altered strain shows protective capacity when administered via the oronasal route compared to parenteral administration of other live attenuated vaccine candidates. Oronasal delivery is more feasible for use in wild animals versus intramuscular injection, and challenge studies showed no difference between delivery routes in protection from the parental ASFV-G strain [33]. Additionally, this vaccine effectively protects both European breeds and native Vietnamese pigs against the circulating Vietnamese field strain, with full protection accomplished by the fourth week post-vaccination [32].

Another gene-deleted strain, namely SY18ΔI226R, was developed in China. This gene deletion has never occurred naturally in circulating field strains, and the gene has been conserved with minor changes in all isolates, indicating it may play a role in the maintenance of virulence. The removal of this gene did not affect replication in cell culture. After inoculation with SY18ΔI226R, study animals remained clinically normal. Following challenge with the parental SY18 strain, all animals in the control group developed clinical signs, and three were euthanized prior to the end of the challenge period. Pigs in the inoculated group survived the challenge, and only two developed low fevers. SY18ΔI226R-immunized pigs shed low numbers of virus on oral swabs, although the shed virus was found to be virulent. This reversion to virulence can occur with live virus vaccines, although it presents a risk for environmental contamination and virus spread [34].

The vaccine candidate ASFV-G-ΔMGF, another live attenuated strain, was evaluated for reversion in vivo. Study animals had transient high body temperatures and moderate inappetence throughout the five passages. ASFV genome was found in all animals after passage five, and all were clinically healthy at the study endpoint. Whole-genome sequencing found that the core genome was stable throughout the study, although a variant virus emerged in passage one and overgrew the parent virus. The authors note that this method of forced animal passage is artificial, and our results must be considered within this context [35].

It is important to note that at the time of review, there are no commercially available vaccines for ASFV in the United States. Additionally, of the vaccines that are licensed in other countries (Vietnam and Philippines), there is currently an inability to differentiate between vaccinated and infected animals, as well as the risk of transmission of the attenuated strain from vaccinated to unexposed animals. As the research into and development of new vaccines continues, these limitations should be considered.

## 2. Swine Farm Biosecurity in the United States

### 2.1. Current Practices: Commercial Operations

Biosecurity in commercial swine production facilities specifically refers to the prevention of the introduction of infectious agents into various production facilities. The term was relatively new to the industry in the early 2000s and has quickly become an everyday consideration for commercial producers at all levels, from boar studs to finishing barns. Any immunologically naïve herd faces greater potential risk in the event of an outbreak, as these animals have rarely been exposed to disease. The biggest concerns for pathogen spread are the introduction of infected animals to new facilities and the concentration of swine facilities in any given area, and as such, regions with a higher density of swine farms must follow stricter biosecurity guidelines [36]. While ASFV has not been detected in the U.S., understanding the current biosecurity practices in context in response to existing infectious diseases provides key insights into the level of preparedness and potential gaps in knowledge that the swine industry must manage in order to combat an emergence of ASFV.

Prepared feed and feed ingredients intended for swine consumption have been shown to harbor pathogens and may transfer diseases to otherwise biosecure premises. When viruses such as ASFV, CSFV, and IAV and their surrogates were used in models mimicking trans-Atlantic and trans-Pacific shipping, some could be recovered from up to 10 of the 11 substrates tested. ASFV remained stable in nine substrates with a half-life of 1.3–2.2 days, shorter than those of other viruses in the study but consistent across ingredients. The ingredients that most frequently supported virus survival included soybean meal, lysine hydrochloride, and complete feed. Conventional soybean meal also had the highest percentage of moisture at 12%. Overall, survival was variable and depended heavily on the characteristics of the feed ingredient and the virus itself. ASFV was the only virus to survive in the control matrix. This may suggest that the feed ingredients protected other viruses from the temperature and humidity fluctuations in the simulated shipping scenario compared to the propylene tube with stock virus as a control [37].

Further research into the half-life of ASFV in shipped feed with more time points demonstrated that the half-lives in most ingredients were longer than those previously seen. With five nodes representing critical points in the feed production and shipping process (ASFV contamination of raw materials, inactivation during processing, recontamination after processing, inactivation during transit, and inactivation during entry into the U.S.), this updated model found that the median annual risk of contaminated shipments of corn entering the U.S. was 2.0%. Other feed ingredients had a lower risk of both initial contamination and recontamination after entry. These other ingredients, such as soybean meal, are typically extruded or solvent extracted prior to import, and these processes reduce the likelihood of virus survival [38].

Extended transport time also correlated with decreased risk, as the longer time in shipment increases the chance of virus inactivation. However, there is a significant knowledge gap in the specifics of ASFV prevalence in feed ingredients due to a lack of surveillance at import. ASFV transmission through infected feed or bedding has not been thoroughly studied, and it may become critical in the fight against the virus moving forward as we look to maintain and improve biosecurity at the farm level [39].

Emphasizing the importance of disease surveillance in feed ingredients, a 2014 PEDV outbreak on an Ohio swine farm was eventually traced back to contaminated feed despite their appropriate biosecurity protocols. The facilities had no visitors within 10 days of the outbreak, and after a thorough review of employee movements, there were no trips to foreign countries. All employees and non-employee contractors were required to follow the company’s biosecurity protocols to enter and exit the facility. Only managers were permitted to move between sites, and these sites were all within the same production flow. These facilities had been free of PRRS or PEDV for over seven years. Supplies such as veterinary materials and semen were assessed for potential contamination, but these items were ruled out as they were not shared between production flows and were disinfected in a fume chamber prior to intake. Additionally, aerosols and droplets likely did not contribute to disease spread in this scenario because affected sites were not located close enough to each other for aerosol transmission to occur. Immediately prior to this outbreak, the company switched to a new supplier for starter pellet feed, and PEDV genetic material was found in this feed after an extensive epidemiological investigation. These starter pellets were disinfected in the same manner as veterinary supplies, eliminating contamination on the surface of the bag but not affecting the pellets inside. Production sites within the same flow that never received the new feed remained PCR negative [40]. Further investigation is needed to determine the true infectivity of feed and appropriate methods of decontamination. Additional research into the potential for ASFV spread through feed and feed ingredients would be a valuable contribution to this field.

In commercial swine production, animals may be moved to different facilities depending on their age and production status. This may involve moving weaned pigs to a grower barn or shipping cull sows out of a breed-to-wean facility, typically within the same “flow” or group of barns encompassing all stages of production. Many companies have dedicated trailers for each facility and will isolate cull trailers completely since these trucks travel to collecting points and livestock auctions, where diseases are prevalent. Scale model livestock trailers were experimentally contaminated with PRRSV-spiked feces to simulate the conditions of vehicles after moving pigs, before undergoing a high-pressure wash like that performed in industrial truck washes. The final step was disinfection with either accelerated hydrogen peroxide (AHP) or glutaraldehyde combined with quaternary ammonium (GQA), depending on the treatment group. Pressure washing alone reduced the quantities of PRRSV genomic material despite the presence of visible fecal material after cleaning, but samples collected from control trailers still resulted in infections when inoculated into live animals. Both disinfectants destroyed viable PRRSV, and samples taken from these trailers after a 15-minute contact time did not infect study animals [41].

In a study examining the efficacy of a peroxygen-based disinfectant, diamond aluminum coupons intended to replicate trailer surfaces were contaminated with PEDV-spiked feces, before being disinfected with 1 of 10 treatments. These treatments used the same disinfectant at different concentrations, temperatures, and contact times, with phosphate buffered saline (PBS) used as a control. Following treatment, environmental samples were taken from each coupon and inoculated into a randomly selected, PEDV-negative barrow. Our results showed that the disinfectant was highly effective in winter-like conditions (4 °C and −10 °C) after a 30-min contact time when applied to low levels of organic matter. This is important in the event of less-than-ideal washing, which may occur, particularly in the winter, as the disinfectant was still effective against PEDV [42]. The combination of effective cleaning and a disinfectant, given the correct contact time, can reduce incidence of disease, particularly of viruses that may be carried on trailers returning to their “home” sites.

Breed-to-wean production sites, where young pigs are born and raised until weaning, when they are transferred to a grower facility, fit a critical niche in the U.S. pork production system. However, they also have a unique role in disease transmission, since young pigs can contract, incubate, and spread diseases like influenza A (IAV) to other farms after weaning. In an effort to understand the IAV burden in suckling pigs, 83 breed-to-wean farms voluntarily shared their farm surveillance data and breeding animal vaccination status. During the six-year period, 23% of the 12,814 samples were RT-PCR positive. Following a season-adjusted multivariate analysis, vaccinating sows against IAV and ensuring gilts were IAV negative at intake were the only measures used to significantly reduce piglet infections at weaning [43].

Regarding specific knowledge of the exact day-to-day biosecurity and management protocols followed by swine producers and the extent to which these producers and their employees understand such recommendations, little scientific literature exists. However, the National Pork Board and Pork Checkoff have published ample materials targeting lay swine producers with audience-specific posters, articles, and other resources in this space. These resources include posters with photographs of good and bad biosecurity practices, articles about steps to take after international travel, and booklets containing basic information on common swine diseases [44]. Both websites also promote the Secure Pork Supply (SPS) program, a collaboration between industry partners, state and federal animal health officials, Iowa State University, and the University of Minnesota. SPS plans aim to allow the continuity of business in the event of FAD incursion, helping producers to prove that their animals are disease free for movement purposes and continue limiting their exposure [45].

### 2.2. Current Practices: Niche and Hobby Farms

Surveys conducted in 2017 reported an average of 226.8 pigs per county fair in Indiana [46], a majority of which were sourced from niche and hobby swine breeders. Minimal scientific literature exists on the exact biosecurity and animal management practices followed by niche and hobby producers on their own premises, including exhibition swine breeders. However, these animals are frequently transported to exhibitions and sales in different areas of the country, and research has been conducted on the contribution of movement patterns to disease spread at these events. It is important to examine the potential role of exhibition swine in ASFV’s spread regardless of ASFV presence, and prior research has outlined this sector’s role.

A study carried out at the 2005 California State Fair surveyed 137 exhibitors showing all livestock species and representing 40 of the state’s 58 counties. Out of the 72 hogs represented in the survey, 90% of market and 100% of breeding animals were to be returned home following the show. Biosecurity precautions were minimal, with 7% of respondents indicating that they did not follow any precautions prior to the exhibition and 10% reporting no major precautions during the exhibition. For all species, the most common guidelines followed were no equipment sharing and no direct physical contact with other animals. In general, low quarantine rates were reported, and 57% of respondents intended to return their animals to a commercial livestock facility. This combination of minimal quarantine (shorter than the incubation period for most illnesses) and return to commercial facilities demonstrated a significant risk of disease outbreak should these animals be exposed to pathogens at the fair [47].

Minnesota ranks in the top 5 pig-producing states in the U.S., yet its exhibition swine population remains largely uncharacterized, like that of other major pig-producing states. A 2012 study collected blood samples from 661 exhibition pigs at slaughter and gathered surveys and interviews from 4-H participants who registered a swine project for their county fairs. Only 9% of fair boards included in the survey required that all animals were sent to slaughter at the conclusion of the show, a measure that can reduce disease spread by preventing the return of animals to production herds. This is particularly important because the population of 4-H swine significantly correlated with the commercial swine population at the county level, with 36% of respondents raising commercial pigs on the same premises and 15 individuals keeping commercial and exhibition swine in the same barn [48]. In contrast, only 13.3% of exhibitors in Ohio and Indiana reported raising commercial and exhibition swine on the same premises [46].

The implications of a show pig being exposed to diseases at a county fair and returning to the same premises or barn with a commercial swine herd are immense. Respondents were presented with a list of biosecurity recommendations and asked to mark them as “important”, “unimportant”, or “unfamiliar.” The top three recommendations rated as important were transport sanitation, sourcing healthy pigs, and separating unhealthy pigs. Participants rated “shower upon entry and exit”, “maintain visitor log”, and “wear mask and gloves” as the least important and were the least familiar with recommendations to provide boot baths and bird-proof barns [48].

Blood sample results showed that seroprevalence for PRRSV in the Minnesota show pig sample was approximately 48%, making exhibition swine a potential reservoir for the disease. However, given the much larger population of commercial animals in Minnesota, the risk of disease spread is likely greater from commercial swine to show animals, rather than the other way around. Unfortunately, it is impossible to evaluate the exact risk posed by exhibition swine due to the transient nature of their population, being purchased in the spring and usually sold or butchered after the county fair in the summer. Youth exhibitors who participated in this study were members of 4-H, and many also held membership in FFA chapters, opening possible routes for education on infectious disease and biosecurity as clubs and chapters prepare for the show season [48].

Swine influenza has been associated with over 400 zoonotic IAV infections from county fairs since 2011, emphasizing the importance of fully understanding exhibition animals’ role in disease spread. While some county fairs are fully terminal and require the shipping of all market animals to slaughter, “jackpot” circuits held earlier in the year are never terminal and facilitate the mixing of animals from different farms for several days at a time before returning home [46]. These circuits may serve as sources of the reassorted IAV strains that later infect humans at county fairs. When jackpot animals were tested starting in April, IAV first appeared as PCR positive in May in the state circuit, then in a June national jackpot, and then throughout the duration of the county fair season from June to October. IAV was detected at significantly higher levels at jackpot shows (76.3%) compared to county fairs (37.8%). Additionally, the virus was detected at 68.9% of state shows and all national shows sampled, regardless of the density of the local swine population. Given the potential for exhibition animals to be infected with or exposed to diseases at shows or fairs, the implementation of appropriate biosecurity measures at these events is important for preventing disease transmission between animals and zoonotic transmission to humans [49].

Representing a significant contrast to commercial operations, niche and hobby swine breeders often host “open barns” at their farms prior to sales, which may also be hosted onsite. Open barn events invite members of the public who are interested in buying pigs, typically for their youth exhibitors or their own breeding stock, to visit the farm and evaluate pigs in person. Farm owners may require the use of boot covers and coveralls, but visitors do not normally shower in and out or limit contact with other swine before or after the event. These pigs are also transported and commingled with other pigs many more times in their lives compared to commercial swine. Some exhibitors reported attending as many as 50 shows before attending their county fair [46].

Although fairs and exhibitions may use signage and announcements to recommend biosecurity measures to exhibitors and members of the public, such recommendations are not always heeded or even viewed as important by individuals in livestock areas. This disregard for biosecurity leads to outbreaks of zoonoses and animal infectious diseases, such as a 2016 H3N2v outbreak linked to agricultural fairs in Ohio and Michigan that caused illness in 18 people, including seven children under the age of five. Every case of reported exposure to swine was recorded at one of the seven fairs, and no person-to-person transmission was found. All individuals fully recovered, and pigs at all seven fairs tested positive for H3N2 [50]. Concurrent with this outbreak, swine exhibitor households participating in eight Michigan fairs were asked to complete surveys exploring their knowledge of and support for various biosecurity recommendations. Despite evidence to the contrary, 90% of respondents perceived their risk of contracting influenza from swine to be low or very low. Exhibitors generally did not support closing barns to the public, holding “distance” swine auctions, or reducing swine shows to under 72 h to minimize the risk of disease transmission, but they indicated overall support for better monitoring of hand wash stations outside of barns [51].

Many county and state fairs require livestock animals to remain on the grounds for over a week depending on the length of the exhibition, regardless of when in that timeframe these animals pass through the show and sale rings. Keeping animals on the grounds and allowing them to interact with each other for extended periods of time exacerbates disease spread, compounded by suppressed immune systems due to the stress of moving into the fairgrounds. This phenomenon led to the “72-h rule”, a suggestion for fair boards to release or sell all swine within 72 h after they arrive on the grounds. One 2014 fair had 3.8% IAV prevalence at 15 h after arrival and 77.5% prevalence after 150 h. Fairs tested in 2018 and 2019 were asked if they released some, all, or none of their pigs before 72 h, and the mean prevalence in fairs releasing all pigs (6.1%) was significantly lower than fairs releasing some pigs (33.2%). It is expected that some animals will arrive carrying diseases, but implementing measures such as the 72-hour rule at a fair level removes the ability for individual exhibitors or families to ignore one particular biosecurity recommendation [52].

Swine exhibitions are an important piece of agricultural fairs, and the life skills that youth learn through participation in such events are invaluable. However, it is undeniable that the exhibition swine population serves as a reservoir for impactful diseases, and their caretakers may not be fully aware of or willing to follow the biosecurity recommendations that have been implemented by commercial swine producers for many years. Moving forward with U.S. foreign animal disease prevention and preparedness, it will be critical to have a comprehensive understanding of niche and hobby swine producers’ and exhibitors’ knowledge of and attitudes toward biosecurity principles.

## 3. Youth Biosecurity Baseline

### 3.1. Perceptions and Understanding

Considering the ongoing HPAI outbreak in 47 states that has affected over 57 million domestic poultry to date [53], it is more important than ever that all youth, especially those involved in agriculture, are well educated in biosecurity principles. Before attempting to offer educational programs, we need to assess the baseline of perceptions of and understanding among youth livestock exhibitors to avoid providing materials that are too far above or below the appropriate level. Most of the scientific literature in this space pertains to biosecurity practices followed at exhibitions rather than a broader understanding of the principles, and it relies heavily on self-reporting through surveys, but these studies evaluate basic knowledge of the concept and recommendations.

Youth agricultural programs such as 4-H and FFA (National FFA [Future Farmers of America] Organization) offer a wide variety of educational activities and livestock exhibition opportunities for young people. Families exhibiting livestock in 4-H classes at the Kansas State Fair were asked to answer a series of questions about their youth exhibitors’ habits before, during, and after the fair. The results covered all species, with some questions being broken down by species. Most exhibitors, i.e., 80.4%, reported checking animals for disease prior to moving them in, while only 11.1% claimed to disinfect footwear before the fair. Among swine exhibitors, 12.9% never disinfected equipment before traveling to a fair, and 30.6% never disinfected shared equipment before using it on their own animals. Nearly 70% of all exhibitors said they do not quarantine animals after returning from an exhibition. The results demonstrate a general lack of understanding of disease transmission and the importance of frequently disinfecting equipment, whether or not it is shared. This survey was, however, limited in its scope due to the exclusion of FFA members and out-of-state exhibitors [54].

The document “Measures to Minimize Influenza Transmission at Swine Exhibitions” (MtM) is published yearly by the CDC in collaboration with human and animal health officials, with recommendations including “clean and disinfect all tack and equipment between shows”, “wash your hands with soap and water when you leave the barn”, and “no food or drink in animal areas.” Nolting et al. aimed to evaluate the impacts of this document on youth swine exhibitors, particularly those exhibiting outside of county fairs alone. Each recommendation put forth in MtM was followed by statements asking about exhibitors’ awareness of, opinion toward, and behavior inspired by the recommendation. Some were already widely followed, such as preventing sick pigs and people from attending shows and disinfecting equipment between shows, but those least likely to be followed were no eating or drinking in barns and isolating animals after they returned from a show. Nearly 80% of exhibitors eat or drink in the barns, and half reported sleeping in animal-adjacent areas [55]. These results indicate some awareness of biosecurity principles and recommendations, but there is resistance to following guidelines, even if they are shown to reduce disease transmission.

### 3.2. Education Efforts

Many educational efforts surrounding biosecurity and disease prevention are geared exclusively toward youth livestock exhibitors and delivered through 4-H and FFA programs. However, there are some external efforts attempting to enhance these programs and evaluate their efficacy. Beginning with 4-H and FFA-based efforts, Minnesota 4-H Animal Science partnered with researchers from the University of Minnesota to develop modules for nine species to be offered at state shows and day camps. The group developed the Biosecure Entry Education Trailer (BEET) as a mobile classroom and taught lessons about disease prevention and biosecurity at each show or day camp. The BEET included a mock barn entryway, and participants were taught the proper methods for crossing a line of separation to enter a biosecure facility. Each participant completed pre- and post-workshop surveys the same day and received an electronic survey six months later to evaluate the long-term impacts of the events. According to the post-workshop surveys, swine exhibitors strongly agreed that they could write a biosecurity plan after the event and saw a 36.5% gain in overall knowledge. Swine and poultry exhibitors were the most comfortable with biosecurity concepts, likely due to the prevalence of outbreaks in their respective industries [56]. The strengths of this format include the hands-on activities, small groups that allow higher-quality interactions with the facilitator, and take-home materials in a biosecurity kit.

One benefit of enhancing youth biosecurity education is the positioning of these youth as “experts” in their communities after the program. This helps to shift power dynamics and empowers young people to lead [57]. California 4-H members recruited by county representatives participated in the “4-H Bio-Security Proficiencies Program”, a multi-week workshop consisting of in-person and distance-learning modules. Following the completion of all modules and evaluations, the students acted as citizen scientists by developing biosecurity improvement plans for the Yuba-Sutter fair. The recommendations in these plans were considered by the fair board, and the students reported which improvements were made, including installing new handwashing stations and removing the mixed species “champions row.” During the year following their county fair, the 4-H members engaged with the UC Davis Veterinary Medicine Extension team to develop educational videos targeting fair leadership, youth exhibitors, and the public. After these community science activities, students were able to identify the changes that they had made, they could watch themselves teach others about biosecurity, and they understood the significance of their roles as citizen scientists [58]. The California program was successful for the same reasons as the BEET team in the sense that it offered students the chance to not only learn about biosecurity but also to apply this knowledge in a very meaningful way to support their community.

Many external education efforts, although not channeled directly through 4-H or FFA programs, still rely heavily on the participation of these members at their county fairs or jackpot shows. Five Ohio counties were chosen for their existing participation in IAV surveillance and greater numbers of enrolled swine exhibitors. During required quality assurance training sessions, research staff added tabletop scenarios designed to demonstrate appropriate animal management before, during, and after exhibitions to reduce IAV transmission. The scenarios were adapted from activities conducted by the Ohio Department of Health during zoonotic disease response training. Exhibitors completed a pre-test, reviewed a short lesson on IAV and disease spread, and broke into small groups to run through an assigned scenario before taking the post-test. The activities helped youth to understand viral transmission between animals and between animals and people, as shown by the score increases of 1.54 and 1.78, respectively, for the relevant questions [59]. This teaching method was effective in its niche but would likely see broader success when combined with a truly hands-on activity like the BEET lesson or community science. Incorporating multiple teaching methods in a One Health approach was the basis for the Healthy Animals/Healthy YOUth program that distributed 120 resource kits and offered training to over 5000 young people in Virginia and Maryland [60].

## 4. Adult Biosecurity Baseline

### 4.1. Perceptions and Understanding

The broad focus of past research in the biosecurity and infectious disease space has been on youth exhibitors and agriculture students, with minimal studies of the baseline knowledge, perceptions, and understanding of adults involved in agriculture. Agricultural educators, 4-H leaders, and youth development and natural resources Extension agents are all likely to interact with youth on a near-daily basis for educational purposes. These groups tend to be aware of zoonotic diseases that they could contract from animals and understand basic precautions like using PPE, but they were less aware of or knowledgeable about their own ability or responsibility to educate others on zoonotic diseases. Respondents were able to identify rabies, salmonella, and ringworm as zoonotic, but incorrectly chose hepatitis and distemper from a list of diseases. Despite a lack of confidence in their ability to teach others about biosecurity or zoonoses, these individuals believe 92% and 87% of the teaching responsibility for these topics should fall to agriculture teachers and Extension agents, respectively. Many respondents report never receiving formal training or education on zoonoses, but they were able to find information easily and knew how to use these resources [61].

With this lack of training and minimal confidence in their own knowledge, it is natural to assume that educators rely on outside curricula. However, some instructors voiced their concerns about a lack of age-appropriate materials for their agriculture classes, instead sourcing community knowledge by bringing in guest speakers and successful producers. In the absence of useful curriculum, these educators turn to their own experiences, and as a result, food animal production lessons, including biosecurity, can be highly varied, even within the same county [62].

There is a significant lack of knowledge about baseline biosecurity and infectious disease perceptions and understanding among adults working in agriculture, despite these adults being expected to teach the next generation. It is important to focus not only on offering youth educational programs in this space but also on tailoring programming to meet the needs of adult agricultural educators to better serve their students and advisees.

### 4.2. Education Efforts

Minimal peer-reviewed literature examining education efforts for adults in agriculture exists. One such study incorporated One Health principles into Train-the-Trainer workshops for 4-H staff, Extension volunteers, agricultural educators, and livestock show officials in Virginia and Maryland, reaching 94 facilitators [60]. This is only a small component of a safe, healthy livestock industry, but little is known about outreach and education efforts available for adults.

## 5. Conclusions

The widespread devastation of ASFV in countries with ongoing epizootics is evident from the previous literature evaluating its impacts. Additionally, there is a significant lack of effective control measures for the reasons described above, including the potential for the reversion of modified live vaccines. Disinfection and decontamination research efforts have produced promising results demonstrating the efficacy of existing commercially available disinfectants against ASFV, although some substrates are challenging to fully decontaminate, and a lack of surveillance for pathogens in feed or bedding at import remains a problem. Biosecurity practices followed in commercial swine operations are well-known, while those followed by niche swine producers are not as well known. Most biosecurity and disease prevention measures known to happen in the niche swine sphere are recorded at the exhibition level, rather than at the farm level, limiting current knowledge of niche swine biosecurity despite the intertwined nature of the swine industry as a whole.

Finally, prior education and outreach efforts have strongly focused on youth exhibitors, with little programming aimed at adults in educational or mentoring roles. These adults have expressed a desire to teach about biosecurity and disease prevention, but self-ratings for knowledge on these subjects were comparatively low. Amid the HPAI epizootic affecting millions of commercial and backyard poultry and with the looming threat of ASFV, it is critical that youth and adults in the exhibition swine space are equally well educated in biosecurity and foreign animal disease preparedness. If these individuals can serve as community educators in their respective social environments, the wider swine industry and the food supply chain it supports will be better prepared for future foreign animal disease incursions.

## Data Availability

Data sharing not applicable.

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
