# Peer review of "African Swine Fever: A Review of Current Disease Management Strategies and Risks Associated with Exhibition Swine in the United States"

_animals, 2023, doi:10.3390/ani13233713_

Round 1

Reviewer 1 Report

Comments and Suggestions for Authors

Hannah Cochran et al, proposed current disease management strategies and risks associated with exhibition swine, especially in the USA scenario. It is helpful for readers to understand the role of exhibition swine in spread of animal infectious diseases, and prevention of these diseases in exhibition swine section. There are some comments on this paper.

1.     Lines 250-252, authors concluded that no vaccines are commercially available despite promising candidates due to concerns for virus spread from vaccinated animals. However, live-attenuated vaccines including ASFV-G-ΔI177L, have been commercially available in Vietnam since 2022. Authors are encouraged to discuss the possibility of ASFV live-attenuated vaccine vaccination in the United States, even for emergent vaccination to prevent ASF introduction, and the possible reasons for forbidding ASFV vaccination practice in the USA might be given as detail as possible.

2.     To validate the use of oral fluids for ASFV detection, the case of ASF Georgia 2007/1 (highly virulent) or ASF Malta’ 78 (moderately virulent) was discussed (Lines 143-155). If low virulent ASFV was involved, is oral fluids suitable for ASFV detection? It might be emphasized since this situation is common in those ASF endemic area.

3.     Lines 57-58, “Feral swine have been reported in 35 states”, a feral swine population distribution map in these 35 states in the United States is encouraged to be present.

4.     Lines 80-82, In a study simulated 30-day trans-Atlantic shipment, the half-life in feed ingredients ranged from 9.6 to 14.2 days. However, in lines 270-273, in another models mimicking trans-Atlantic and trans-Pacific shipping, ASFV remained stable in 9 substrates(ingredients) with a half-life of 1.3-2.2 days. The former half-life of ASFV is longer than that in the later(9.6 to 14.2 days & 1.3-2.2 days), authors should discuss what make this difference, avoiding to puzzle readers.

5.     4-H or FFA programs should be given a brief explanation.

Author Response

Please see attachment. Thank you for your consideration.

Reviewer 2 Report

Comments and Suggestions for Authors

The review titled 'African Swine Fever: A Review of Current Disease Management Strategies and Risks Associated with Exhibition Swine' by Cochran et al. aims to explore the potential spread of ASFV in the context of swine exhibitions and emphasizes the crucial role of public education in mitigating this risk.

While the subject matter is pertinent considering the global spread of ASFV, the manuscript requires significant revisions before publication. Several sections are addressed superficially, demanding enhancement for improved relevance and readability. Therefore, the recommendation is for major revisions and resubmission.

Specific comments:

1. To begin with, there is a need for restructuring and refining the flow between endemicity, transmission, and detection. Particularly, section 1.1 discusses how humans can reintroduce diseases without introducing the issue of transmission until section 1.2, potentially causing confusion among readers unfamiliar with ASFV transmission. It is strongly recommended that the authors reevaluate the sequence of these sections for a more coherent flow.

2. In section 1.2, the shift from discussing transmission to detection appears abrupt, leading to confusion for readers. It becomes evident that the authors are addressing detection only later in the paragraph. To enhance the readability of these sections, it is essential for the authors to provide clear introductory statements at the beginning of each paragraph. These statements should outline the specific evidence to be discussed within the paragraph, providing readers with a roadmap for the content. Additionally, a comprehensive introduction to the commonly used detection methods is necessary. This introduction should include detailed explanations of the methods and established procedures, all of which need to be properly cited to ensure accuracy and credibility.

3. Additionally, there are issues with formatting and citation. Many references are not properly formatted, requiring correction. 

4. The introduction of new vaccines, especially those licensed in Vietnam and the Philippines, should be supported by recent publications instead of relying on older references. 

5. At line 84, the acronym PEDV is introduced for the first time; hence, it is imperative to provide its full definition to ensure clarity for readers.

6. Furthermore, certain sentences lack clarity, such as in line 88 where the importation or introduction of something is not specified. 

7. Ambiguous terms like 'treat' in line 93 should be replaced with clearer alternatives like 'contaminate.' 

8. The authors must also provide citations for guidelines mentioned in line 100.

9. In terms of content, section 1.3 lacks an explanation of the term 'survivability,' and the authors should define it for better comprehension. 

10. The mention of four concentrations in line 175 needs elaboration. 

11. The formatting of italics in the text requires correction; certain words that should be in italics are currently not formatted accordingly.

12. Additionally, the authors should clarify the meaning of 'further titers' in line 199. 

13. In line 200, the authors should consider 'at all time points assessed within the study'.

14. Lines 201-209: The discussion on meat curing and drying processes contradicts itself, as it states that these methods can inactivate viruses while citing evidence of live ASFV in cured meats. This inconsistency needs to be addressed. The survivability of ASFV has been demonstrated by other research groups. The authors should cite relevant works, such as the study with DOI: 10.3390/pathogens9110977, to provide comprehensive and credible support for their statements. Furthermore, the presence of ASFV genome versus live virus capable of infecting animals should be clearly differentiated.

15. Regarding biosecurity, the authors overlook current practices in China, the largest pork producer globally. A section detailing contemporary methods of herd management in the context of ASFV, such as partial depopulation and partitioning, should be included (doi: 10.3389/fvets.2021.812876). 

16. The statement about the availability of commercially available vaccines is inaccurate (Lines 219 and 250). Vaccines are now licensed in Vietnam and the Philippines. 

17. In line 221, the statement is not entirely accurate. There exists a diverse array of vaccines, and certain types, such as modified live viruses (now licensed), have proven highly effective against specific strains they were designed to combat.

18. The authors should clarify the complexities of live virus vaccines, including safety concerns and the differentiation between vaccinated and infected animals. Mention of recent studies on reversion to virulence and the ASFV-G-dI177L vaccine strain's ability to transmit to non-vaccinated animals is essential for comprehensive coverage. The ASFV-dMGF vaccine strain that has been implemented in Vietnam and the Philippines. Additionally, a study conducted by FLI demonstrated the potential of the ASFV-dMGF vaccine strain to revert to virulence, emphasizing the need for comprehensive coverage of vaccine research outcomes. Subunit vaccines, currently under development by various research groups, pose unique challenges due to the intricate nature of ASFV. Their complexity makes them technically demanding to create.

19. Furthermore, the manuscript should address ASFV spread through feed in a dedicated section, including recent publications on this topic (Lines 278-290). 

20. Lines 291-294: authors should also include the research on thermal inactivation of feed and the development of feed additives that can mitigate the risk of ASFV introduction through feed.

21. Lines 317-332: Research on trucks involved in ASFV spread should also be introduced. 

22. In section 2.1, the authors must specify their use of insights from other porcine viruses to understand potential ASFV transmission in the absence of ASFV in the US. 

23. Additionally, the review should encompass biosecurity measures for show/fair organizers and provide recommendations beyond public education to mitigate ASFV spread in exhibition swine.

Comments on the Quality of English Language

In terms of the English language, the manuscript is satisfactory. However, there are formatting issues, particularly regarding the correct use of italics, that need to be addressed and corrected.

Author Response

Please see the attachment. Thank you for your consideration.

Round 2

Reviewer 1 Report

Comments and Suggestions for Authors

All my comments have been placed down, the paper can be accepted.

Author Response

Thank you for your comments and time spent reviewing.

Reviewer 2 Report

Comments and Suggestions for Authors

It is evident that the authors have made only superficial amendments to the manuscript without seriously considering the comments provided. A key objective of writing a review article is to synthesize the most pertinent and reliable information available in literature to produce a comprehensive, up-to-date and reader-friendly overview for a specific research area. Therefore, it is crucial to offer unbiased background information and provide clear recommendations for future research or measures. If the authors have made any recommendations, these have not been effectively communicated in the text, and they need to emphasize and consolidate these at the end of the manuscript to enhance reader comprehension.

Addressing the previous review, the mention of 'close contacts between humans and animals' in line 38 could potentially mislead readers into considering ASFV as a zoonotic virus. The authors should carefully rephrase this to avoid confusion.

Reference 1 is inappropriate for citing the vaccines licensed in Vietnam and the Philippines, especially since these vaccines have been licensed this year, and the vaccine used in the Philippines does not align with the one studied in reference 1. Authors must reference the relevant literature to provide accurate and updated information.

Additionally, the manuscript lacks a comprehensive introduction to commonly used laboratory detection methods for ASFV, particularly those employed by reference laboratories. Proper citation of these methods is essential.

When discussing the vaccine strain ASFV-G-ΔI177L, it is imperative to highlight not only its advantages but also its limitations. This vaccine strain has the capacity to transmit between vaccinated and naïve animals and is not DIVA compliant, meaning vaccinated animals cannot be differentiated from infected ones. All technologies presented in the manuscript should be assessed critically, covering both benefits and drawbacks.

Italicization of terms that require it has been neglected, and the acronym FFA must be defined in full to enhance reader understanding.

Concerning lines 218-219, it is doubtful that the curing process can completely destroy the virus if it takes 140 days for inactivation. This duration is comparable to survival in hay and straw and worse than survival in water and soil, as cited in lines 223-226. This inconsistency needs clarification and further explanation.

In Section 2 on Swine Farm Biosecurity, although the review is submitted under this category for the United States, readers would benefit from a more comprehensive and up-to-date perspective. Therefore, the authors should include recent research on newer methods of herd management which would be relevant to progressing swine farm biosecurity in the United States.

Furthermore, the manuscript overlooks recent publications on ASFV spread through feed, an essential topic for an updated review. Additionally, research on trucks involved in ASFV transmission should be incorporated to provide a holistic view of the subject matter. A comprehensive review should offer the latest information on the developments in the areas discussed, ensuring the content remains current and informative for readers. 

In its present form, the review falls short and necessitates significant revisions to meet the required standards for publication. I trust the authors will thoughtfully consider the feedback provided and make the necessary revisions, respecting everyone's time and efforts involved in the review process.

Comments on the Quality of English Language

Some minor English editing is needed.
